# Influenza Vaccination and Cardiovascular Outcomes in Patients with Coronary Artery Diseases: A Placebo-Controlled Randomized Study, IVCAD

**DOI:** 10.3390/vaccines13050472

**Published:** 2025-04-27

**Authors:** Mohammadmoein Dehesh, Sharareh Gholamin, Seyed-Mostafa Razavi, Ali Eskandari, Hossein Vakili, Mohammad Rahnavardi Azari, Yunzhi Wang, Ethan K. Gough, Maryam Keshtkar-Jahromi

**Affiliations:** 1Division of Infectious Diseases, Department of Medicine, Johns Hopkins University School of Medicine, Baltimore, MD 21224, USA; mohammadmoein.dehesh@ucf.edu; 2Ciccarone Center for the Prevention of Cardiovascular Disease, Johns Hopkins University, Baltimore, MD 21187, USA; 3Department of Radiation Oncology, City of Hope National Medical Center, Duarte, CA 91010, USA; sharareh.gholamin@gmail.com; 4Department of Neurology, University of South Dakota Sanford School of Medicine, Sioux Falls, SD 57069, USA; mostafa_razavi@yahoo.com; 5Department of Radiology, University of Iowa, Iowa City, IA 52242, USA; mralieskandari@yahoo.com; 6Cardiovascular Research Center, Shahid Beheshti University of Medical Sciences, Tehran 1985717443, Iran; hosavak@yahoo.com; 7Department of Cardiothoracic Surgery, Liverpool Hospital, Liverpool, NSW 2170, Australia; drazaricts@gmail.com; 8Epidemiology and Data Management Center, Johns Hopkins School of Medicine Biostatistics, Baltimore, MD 21205, USA; ywang694@jhmi.edu; 9Department of International Health, Human Nutrition Program, Johns Hopkins, Bloomberg School of Public Health, Baltimore, MD 21205, USA; egough1@jhu.edu

**Keywords:** coronary artery disease (CAD), influenza vaccine, cardiovascular outcome, antibody

## Abstract

**Background/Objectives**: Influenza infection is associated with cardiovascular morbidity and mortality; however, the effect of influenza vaccination on cardiovascular outcomes is not fully understood. This clinical trial aimed to investigate the correlation between cardiovascular outcomes and influenza vaccine (FluVac) in coronary artery disease (CAD) subjects. **Methods**: This was a randomized single-blinded placebo-controlled trial. Enrolled CAD subjects received 0.5 mL of 2007–2008 trivalent FluVac (15 µg hemagglutinin of each of Solomon Islands/3/2006 (H1N1), Wisconsin/67/2005 (H3N2), and Malaysia/2506/2004 (B)). The subjects were followed up at 1 month (hemagglutinin (HA) antibody titers) and at 12 months post-vaccination for evaluation of outcomes (influenza-like episodes, acute coronary syndrome (ACS), myocardial infarction (MI), coronary revascularization, and death). **Results**: In total, 278 eligible CAD subjects were randomized to receive either FluVac (*n* = 137) or a placebo (*n* = 141), of which consequently 131 and 135 subjects completed the study. Cardiovascular deaths (3/131 [2.29%] vs. 3/135 [2.22%]) and all-cause deaths (4/131 [3.05%] vs. 4/135 [2.96%]) were similar in both groups. Adverse cardiovascular events, including ACS, MI, and coronary revascularization, were less frequent in the vaccine group but did not reach statistical significance. The magnitude of the antibody change and serologic response (≥4-fold HI titer rise) of all three antibodies were significantly higher in the vaccine group compared to the placebo but did not correlate with cardiovascular outcomes in the FluVac group. **Conclusions**: The influenza vaccine may improve cardiovascular outcomes, though this improvement is not correlated with post-vaccination antibody titers. Despite the controversy, influenza vaccination is recommended in the CAD population (clinicaltrials.gov; NCT00607178).

## 1. Introduction

Coronary artery disease (CAD) is the leading cause of death, with more than USD 400 billion in annual direct and indirect costs in the United States [1]. Most individuals who experience a myocardial infarction (MI) are reported to have at least one cardiovascular risk factor before the event [2,3]. It is well established that moving toward ideal cardiovascular health not only prevents cardiovascular adverse events but also is associated with decreases in heart failure, cancer, depression, and cognitive impairment [4,5].

An association between influenza vaccination and improved cardiovascular mortality has been suggested [6,7,8]; however, the mechanisms remain unknown [9,10,11]. Several clinical trials have evaluated the correlation between influenza vaccination and cardiovascular outcomes; this correlation is also still controversial [12,13,14,15].

In this trial, we evaluated the correlation between the 2007–2008 influenza vaccine (FluVac) and cardiovascular outcomes in CAD patients. In phase 1 (previously reported), we evaluated the antibody (Ab) response to influenza vaccine in CAD patients, which was comparable to the healthy controls [16]. In phase 2 (current report), we evaluated the cardiovascular outcomes 12 months after vaccination in vaccine recipients in comparison with the placebo controls. The delay in reporting the result was due to our time limitations.

## 2. Materials and Methods

### 2.1. Study Design and Ethics

This randomized, placebo-controlled, and single-blinded clinical trial (IVCAD) was performed at Shahid Modarres Medical Research Center, Shahid Beheshti University of Medical Sciences, Tehran, Iran. Participants were enrolled from January to August 2008.

IVCAD was conducted following the principles of the Declaration of Helsinki and received approval from the ethics committee of Shahid Beheshti University, Tehran, Iran (Study# SBMU-86-03-105-5433A; date: 01/05/2008). Before enrollment, all participants provided informed written consent. The trial was registered at clinicaltrials.gov (NCT00607178).

### 2.2. Participants

We screened 300 CAD patients, of which 278 subjects enrolled in the trial. The eligibility criteria to be enrolled in the study were age ≥25 years and having been diagnosed with stable angina or confirmed coronary artery stenosis by angiography or recent MI after recovering from the acute phase. MI was diagnosed via either a pathological finding of an acute MI [17], or a change in biochemical markers (rise and gradual fall (troponin T), or a more rapid rise and fall (CK-MB)) of myocardial necrosis with at least one of the following: ischemic symptoms, development of pathologic Q waves or ST segment elevation or depression on the electrocardiogram (ECG), or coronary artery intervention (e.g., coronary angioplasty). Twenty-two subjects failed the screening, since they met the exclusion criteria. We excluded subjects who had any acute disease, unstable angina, chronic kidney diseases, chronic liver diseases, immunosuppression (e.g., transplantation, Human Immunodeficiency Virus (HIV)), confirmed history of active malignancy, inoculation with influenza vaccine or severe influenza infections requiring hospitalization within the past five years, any psychological illness that could interfere with the regular follow-up, congestive heart failure (NYHA III/IV and/or Killip class IV [18]), and contraindications of vaccine inoculation (e.g., egg allergy). To determine the basic characteristics, a detailed medical history was taken from all subjects, and they underwent a complete echocardiography assessment. Additionally, angina severity and coronary artery stenosis were assessed using the Seattle Angina Questionnaire (SAQ) [19] and the Modified Gensini Score (MGS) [20], respectively. The SAQ serves as a reliable tool for assessing coronary artery disease surveillance. It encompasses five key factors: physical limitations, angina stability, angina frequency, treatment satisfaction, and disease perception. These criteria are employed to evaluate clinical changes and cardiovascular outcomes [19]. The MGS is a scale used to estimate the degree of stenosis in coronary arteries. This grading system assigns scores to the main vessels and their branches based on the severity of obstruction (<50%, 50–74%, 75–99%, and 100%) observed in coronary angiography. The MGS was scored for all subjects by the same cardiologist. By combining the individual scores, final scores ranging from 0 to 20 were calculated. Higher scores are indicative of a higher degree of severity in coronary artery disease [20].

### 2.3. Randomization, Blinding, and Trial Intervention

Eligible subjects were randomized by the principal investigator to receive either the influenza vaccine or a placebo in a 1:1 ratio using a computer-generated sequence. The participants were blinded to group assignment. In total, 137 subjects were assigned to the vaccine recipient group (CAD-FluVac), and 141 subjects were assigned to receive the placebo (CAD-Placebo). The influenza vaccine (FluVac-trivalent influenza vaccine, Solvay Pharma) or placebo was administered as a single 0.5 mL dose into the deltoid muscle. The vaccine contained 15 µg hemagglutinin of each of the three strains, namely Solomon Islands/3/2006 (H1N1), Wisconsin/67/2005 (H3N2), and Malaysia/2506/2004 (B), according to the World Health Organization guidelines for the anti-influenza vaccination campaign of 2007–2008. The control subjects received one 0.5 mL dose of distilled water into the deltoid muscle. All subjects were continued on their standard CAD treatments with no change.

### 2.4. Antibody (Ab) Response

The subjects were evaluated before the intervention and at a 1-month follow-up for a humoral immune response to the vaccine. During the first follow-up visit at one month, Ab titers were measured using a hemagglutination inhibition (HI) test [21], which utilized hemagglutinin antigens representing the virus strains contained in the vaccine. The test involved two-fold dilutions of the serum, ranging from 1:10 to 1:1280. Titers lower than 1:10 were recorded as 1:5. The result of the antibody response was published separately [16]. A protective titer or seroconversion was defined as a ≥4-fold rise in the HI titer.

### 2.5. Outcomes

Subjects were followed up at 12 months via phone call for outcome evaluation. During this final follow-up, subjects were questioned about MI, acute coronary syndrome (ACS), coronary artery bypass graft surgery (CABG), percutaneous intervention (PCI), and the number of cold episodes since their last visit. Additionally, we recorded whether the patient had been reported as deceased (at any time point during the study) through their emergency contact or access to the medical record if available, and we investigated whether death was due to cardiovascular cause or other non-cardiac etiologies. Six subjects in each group were lost to follow-up. Losses were due to changes in the contact information of both the subject and the emergency contact.

### 2.6. Sample Size Estimation and Statistical Analysis

The sample size was estimated to detect a 10% difference in protection rate with an α error of 0.05 and a study power of at least 0.7. Therefore, we aimed to recruit about 300 subjects in this study.

### 2.7. Analysis

The demographic and clinical characteristics at enrollment were characterized with descriptive statistics (frequency and percentage or mean and standard deviation (SD) for categorical and continuous variables, respectively).

Primary and secondary binary outcomes were compared between randomized arms using logistic regression to estimate the odds ratios (ORs) and 95% confidence intervals (95% CIs). Secondary event count outcomes (acute coronary syndrome and cold episodes) were compared between the randomized arm using generalized linear regression with a Poisson distribution, a log-link, and an offset for person-months of follow-up to estimate incidence rate ratios (IRRs) and 95% CIs.

We utilized Fisher’s exact test to determine differences in protective hemagglutination inhibition (HI) titers between the Influvac and placebo groups at the 1-month follow-up. Kruskal–Wallis tests were used to assess differences in the magnitude of antibody titers between groups. These analyses were repeated at the 12-month follow-up for respondents who received Influvac to examine differences between patients who did and did not experience a cardiovascular event during follow-up.

Statistical significance was determined at α = 0.05. All statistical analyses were conducted using R version 4.0.0.

## 3. Results

### 3.1. Baseline Characteristics

Between January and August 2008, 278 CAD subjects (mean age 54.73 [SD 15.4], female 33.5%) were enrolled and randomized into the CAD-FluVac (*n* = 137) or CAD-Placebo (*n* = 141) group. Six subjects (4%) in each group were lost to follow-up (Figure 1).

The demographics and baseline conditions between the two groups, including background diseases, smoking history, and body mass index (BMI), were comparable. In addition, the cardiologic characteristics, including the echo finding, SAQ, and CAD management of subjects, were similar between the two groups (Table 1 and Table 2).

### 3.2. Twelve-Month Outcome

Cardiovascular death was not significantly different between the CAD-FluVac and CAD-Placebo groups (three (2.29%) vs. three (2.22%)). Similarly, the all-cause mortality did not differ between the two groups (four (3.05%) vs. four (2.96%)). However, ACS, MI, and cardiovascular interventions were less frequent in the CAD-FluVac than the CAD-Placebo group but did not reach statistical significance (Figure 2).

The numbers of ACS outcomes and cold episodes were also separately analyzed, showing fewer episodes in CAD-FluVac, but these did not reach statistical significance (IRR: 0.69 [0.37–1.124], P: 0.22) (Figure 3).

### 3.3. Antibody Response

The antibody response in the two groups is depicted in Table 3. The protective titer of anti-H1N1 antibody was significantly higher in the CAD-FluVac vs. the CAD-Placebo group (53% vs. 46%, *p*-value < 0.0001). The magnitude of the change and serologic response (≥4-fold HI titer rise) of all three antibodies were significantly higher in the CAD-FluVac group (Table 3).

We also conducted a subgroup analysis within the CAD-FluVac group to evaluate the correlation between the antibody titer and the cardiovascular outcomes. There was no statistically significant difference in antibody titers between the vaccinated participants who had cardiovascular events and those who did not (Table 4).

## 4. Discussion

Influenza is a serious illness that can impact high-risk populations, causing high mortality and morbidity, with a subsequent significant impact on healthcare resources [22,23,24]. This is more prominent in the elderly and individuals with chronic diseases [25,26]. Influenza infection has a higher impact on CVD patients and can lead to MI and HF, leading to hospitalizations and readmissions due to subsequent morbidities [8,27,28,29].

Influenza vaccination has been shown to be beneficial in patients with cardiac diseases. The relationship between influenza vaccination and cardiovascular outcome has been reported in multiple observational trials in the past [30,31,32,33,34,35], but few clinical trials have evaluated the correlation. The increased cardiovascular risk after influenza infection can be explained by several mechanisms, including the destabilization of atherosclerotic plaque, increased macrophages in the circulation, proinflammatory molecules, such as tumor necrosis factor (TNF)-like weak inducer of apoptosis (TWEAK), and sympathetic activation [3,36,37,38,39,40,41,42,43]. However, it remains controversial whether the influenza vaccine can prevent subsequent cardiovascular events or improve cardiovascular outcomes.

IVCAD was originally designed in 2008 to evaluate this correlation; the data have been used in multiple meta-analyses in 2013, 2015, and most recently 2023 [30,34,35], despite the time limitations in publishing. We decided to publish this full report, due to continued discussions and remaining controversies in this field.

As our data show, IVCAD did not show any statistically significant correlation between influenza vaccination and 12-month cardiovascular/all-cause mortality. However, influenza vaccination was associated with a 27% reduction in the odds of cardiovascular adverse events (ACS, MI, cardiovascular interventions), although this did not reach statistical significance.

The results of IVCAD are almost comparable to a recently performed large multicentric placebo-controlled clinical trial conducted by Loeb et al. They enrolled 5129 heart failure subjects in Asia, the Middle East, and Africa and did not find any significant improvement in non-fatal MI, non-fatal stroke, heart failure hospitalization, all-cause death, or cardiovascular death during a 2.3-year follow-up period in the vaccinated group compared to the placebo group. Interestingly, there were clear reductions in cardiac death, all-cause death, and pneumonia during peak influenza circulation in the vaccinated group, suggesting benefits [15].

Another trial, conducted by Phrommintikul et al., recruited 439 subjects admitted for ACS during the 2007–2008 season. The trial reported a significant reduction in major adverse cardiovascular outcomes (death, hospitalization for ACS, heart failure, and stroke) at the 12-month follow-up after influenza vaccination but did not show any significant improvement in cardiovascular death [14].

Two recent meta-analyses by Gupta et al. [44] and Omidi et al. [45], in 2023, included data from several existing clinical trials. The included trials did not involve similar populations, and some lacked a placebo control group. All these trials concluded that influenza vaccination has protective effects against cardiovascular adverse events and suggested further research to elucidate the precise effects and mechanisms of influenza vaccination on cardiovascular outcomes, while considering vaccination for CAD patients [45].

The IVCAD study also did not demonstrate a significant difference in the antibody response to the influenza vaccine in CAD patients in subpopulations with and without cardiovascular events. In other words, the antibody response does not seem to be the only mechanism involved in the influenza vaccine’s effects on CV outcome. To the best of our knowledge, there are no other existing parallel data investigating CV outcomes based on antibody response to the influenza vaccine. In a prior publication in the IVCAD study population, the antibody response in CAD patients was comparable to that in the healthy controls [16]. A recent similar clinical trial investigated antibody response to high-dose trivalent compared to standard quadrivalent influenza vaccine in subjects with heart failure or myocardial infarction. A more robust humoral response was observed in the high-dose vaccine, but the antibody response had no association with mortality or cardiopulmonary hospitalization [46].

Inflammation is well established to be one of the main baseline conditions resulting in CV events, and multiple studies have shown a protective correlation of the influenza vaccine, preventing influenza infection and reducing systemic inflammation [11,47,48]. Thus, the vaccine has a direct protective effect by preventing influenza infection primarily via an antibody response [49]. On the other hand, the influenza vaccine has been shown to cause a non-influenza seasonal reduction in cardiovascular mortality [12,13], which could not be well investigated in this trial due to the small sample size. These findings suggest the non-antibody-related or nonspecific effect of the influenza vaccine on reducing cardiovascular events. Immune system modulation is a possible explanation. Since a regulated immune system is associated with cardiovascular health, the influenza vaccine plays the role of modifying plaque instability and pre-/post-infarction inflammation [50,51,52,53]. Moreover, the cellular immune response following the influenza vaccine might have a more important role than humoral immunity [54,55]. Although the exact mechanisms and pathways require further investigations, the preliminary data support the idea that future research should focus on parallel pathways including cellular immunity and cytokine regulation in response to influenza vaccination to identify this correlation.

The IVCAD study has some limitations to be considered in parallel with the results. The study had a small sample size, and its results should be interpreted cautiously. Our sample size was calculated to have 70% power to detect a 10% difference in protection rate between treatment arms at α = 0.05 [7]; it may have been underpowered for detecting differences in outcomes due to the low outcome event rates in the placebo arm in our study sample (<10% for most outcomes). Given the low event rates we observed, the absolute risk differences between the treatment arms were generally <=2%, and were thus below our target effect size. Larger trials are needed to more precisely estimate the effect for this vaccine. In contrast to some more recently designed trials, we did not record the seasonal outcomes, which might have provided more detailed and related cardiovascular outcomes during peak virus circulation. Also, due to the low number of outcome events, a reliable subgroup analysis to explore seasonal variations was not feasible [56]. The follow-up period for the current study was 12 months, which is similar to that in most of the other studies, but a longer follow-up period with serial annual vaccinations would have been preferred. Moreover, the outcomes were cumulative and could have happened any time after vaccination until 12 months post-vaccination. A few clinical trials, such as the study by Loeb in 2022, which followed subjects for 2.3 years with annual vaccination [15], and the study by Modin in 2018, with a follow-up period of 3.7 years and variable vaccine coverage [31], reported similar results demonstrating the cardiovascular protective effects of the influenza vaccine. IVCAD was also a single-blinded study (due to technical limitations in its design), which is not the preferred methodology in clinical trials and could have impacted its validity.

## 5. Conclusions

Our study shows that influenza vaccination is associated with fewer cardiovascular events in the CAD population; however, the antibody titer was not correlated with cardiovascular outcomes. This may be explained by potential non-antibody-mediated molecular and cellular immune regulatory pathways for this observation. Further research to investigate other non-antibody immune correlates is recommended. Based on the existing literature, influenza vaccination is highly recommended for CAD populations, with potential benefit in improving their outcomes. Further research, especially large double-blinded controlled clinical trials with a longer follow-up period focused on non-antibody-mediated correlates, is needed to elucidate the precise effect of vaccination on cardiovascular events and to eliminate confounding effects.

## Figures and Tables

**Figure 1 vaccines-13-00472-f001:**
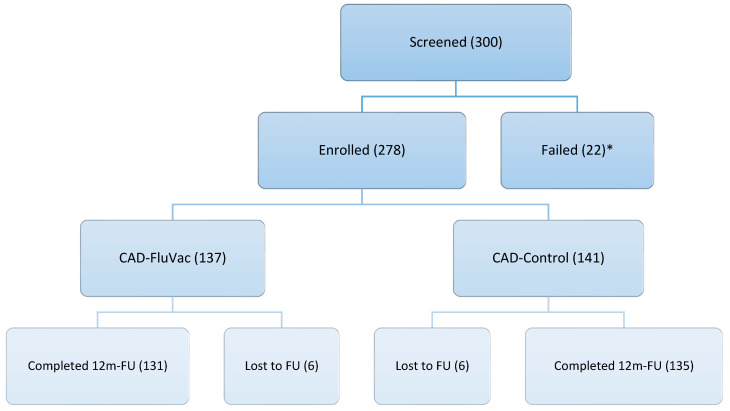
Study flow diagram. * Screen failures were due to exclusion criteria. CAD: coronary artery disease; 12 m: 12 months; FU: follow-up.

**Figure 2 vaccines-13-00472-f002:**
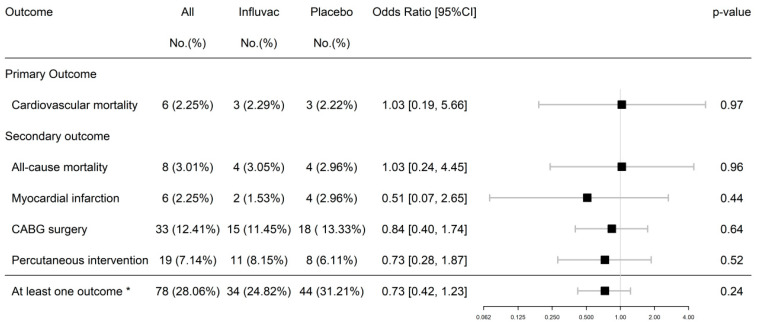
Twelve-month cardiovascular outcomes (yes/no) in subjects with CAD: comparison of FluVac vs. placebo group. * At least one outcome (acute coronary syndrome, myocardial infarction, coronary artery bypass graft (CABG), percutaneous intervention, or death). Analysis: Logistic regression used to estimate incidence rate ratios (IRRs) and 95% confidence intervals (95% CIs).

**Figure 3 vaccines-13-00472-f003:**
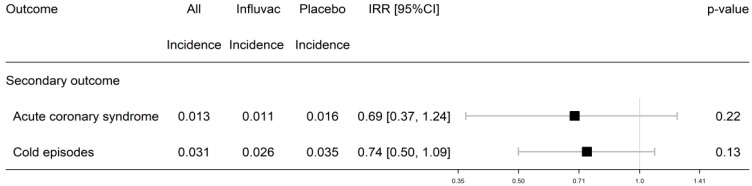
Twelve-month cardiovascular outcomes (numerical) in subjects with CAD: comparison of FluVac vs. placebo group. Analysis: Generalized linear regression with a Poisson distribution, a log-link, and an offset for person-months of follow-up used to estimate incidence rate ratios (IRRs) and 95% confidence intervals (95% CIs).

**Table 1 vaccines-13-00472-t001:** Demographic features and background history of participants.

	All (*n* = 278)	CAD-FluVac (*n* = 137)	CAD-Placebo (*n* = 141)
Mean age, mean years (SD)	54.73 (9.08)	54.53 (9.21)	54.93 (8.98)
Female *n* (%)	93 (33.5)	45 (32.8)	48 (34.0)
Mean BMI (SD)	27.69 (4.47)	27.63 (4.48)	27.75 (4.48)
Background diseases (History)			
Diabetes mellitus *n* (%)	75 (27.0)	35 (25.5)	40 (28.4)
Hypertension *n* (%)	231 (83.1)	115 (83.9)	116 (82.3)
Hyperlipidemia *n* (%)	155 (55.8)	77 (56.2)	78 (55.3)
Daily aspirin *n* (%)	254 (91.4)	128 (93.4)	126 (89.4)
Daily multivitamin *n* (%)	15 (5.4)	9 (6.6)	6 (4.3)
Smoking history, pack/year (SD)	10.66 (21.11)	10.04 (20.65)	11.26 (21.61)
Exercise * *n* (%)	138 (49.8)	72 (52.6)	66 (47.1)
Family history of IHD *n* (%)	133 (47.8)	61 (44.5)	72 (51.1)

* Exercise was defined as minimum of 15 min physical activity including walking daily for at least 4 days per week. BMI: body mass index, IHD: ischemic heart disease.

**Table 2 vaccines-13-00472-t002:** Baseline cardiologic features in participants.

	All	CAD-FluVac	CAD-Placebo
Echocardiography findings	*n* = 180	*n* = 88	*n* = 92
Estimated EF mean (SD)	52.81 (10.09)	52.55 (10.46)	53.07 (9.78)
LV systolic dysfunction *n* (%)	32 (17.8)	17 (19.3)	15 (16.3)
LV diastolic dysfunction *n* (%)	75 (41.7)	35 (39.8)	40 (43.5)
Septal wall akinesia *n* (%)	76 (42.2)	36 (40.9)	40 (43.5)
Myocardial aneurysm *n* (%)	4 (2.2)	2 (2.3)	2 (2.2)
Heart valve abnormalities *n* (%)	55 (30.6)	21 (23.9)	34 (37.0)
Angina severity (SAQ)	*n* = 278	*n* = 137	*n* = 141
Physical limitation mean (SD)	71.81 (22.69)	71.00 (25.24)	72.58 (20.04)
Angina stability mean (SD)	37.26 (30.86)	36.52 (31.16)	37.98 (30.67)
Angina frequency mean (SD)	66.27 (21.93)	65.33 (21.78)	67.16 (22.11)
Treatment satisfaction mean (SD)	75.52 (20.35)	74.69 (21.37)	76.32 (19.37)
Disease perception mean (SD)	56.63 (26.41)	54.86 (26.44)	58.33 (26.37)
Gensini Score mean (SD)	8.74 (5.34)	7.97 (5.03)	9.50 (5.54)
CAD management	*n* = 278	*n* = 137	*n* = 141
PCI *n* (%)	105 (37.8)	47 (34.3)	58 (41.1)
CABG *n* (%)	46 (16.5)	21 (15.3)	25 (17.7)
Medical *n* (%)	270 (97.1)	131 (95.6)	139 (98.6)

EF: ejection fraction, LV: left ventricle, SAQ: Seattle Angina Questionnaire, CAD: coronary artery disease, PCI: percutaneous intervention, CABG: coronary artery bypass graft.

**Table 3 vaccines-13-00472-t003:** Pre- and post-vaccination Ab titers (Month 1) in subjects with CAD: FluVac vs. placebo group.

Outcome	CAD-Placebo (*n* = 141)	CAD-FluVac (*n* = 137)	*p*-Value
Antibody A (Solomon Islands/3/2006 (H1N1))			
Protective (≥1:40) pre-vaccination, *n* (%)	80 (47.62%)	88 (52.38)	0.12
Protective (≥1:40) post-vaccination, *n* (%)	109 (46.98%)	123 (53.02%)	<0.0001
Magnitude of change *, ×fold, mean (SD)	16.30 (66.65)	57.92 (159.81)	<0.0001
Serologic response (≥4-fold rise), *n* (%)	50 (35.46%)	91 (64.54%)	<0.0001
Antibody B (Wisconsin/67/2005 (H3N2))			
Protective (≥1:40) pre-vaccination, *n* (%)	113 (49.34%)	116 (50.66%)	0.10
Protective (≥1:40) post-vaccination, *n* (%)	126 (50.20%)	125 (49.80%)	0.14
Magnitude of change, ×fold, mean (SD)	6.41 (21.59)	28.48 (114.76)	<0.0001
Serologic response (≥4-fold rise), *n* (%)	33 (28.45%)	83 (71.55%)	<0.0001
Antibody C (Malaysia/2506/2004)			
Protective (≥1:40) pre-vaccination, *n* (%)	116 (49.79%)	117 (50.21%)	0.18
Protective (≥1:40) post-vaccination, *n* (%)	135 (51.72%)	126 (48.27%)	0.24
Magnitude of change, ×fold, mean (SD)	6.60 (28.18)	12.45 (56.60)	<0.0001
Serologic response (≥4-fold rise), *n* (%)	64 (39.26%)	100 (60.98%)	<0.0001

* After/before, CAD: coronary artery disease, IQR: Inter Quartile Range. Analysis: Fisher’s exact test used for protective level of antibody and serologic response and Kruskal–Wallis test used for magnitude of change.

**Table 4 vaccines-13-00472-t004:** Correlation between antibody response and 12-month cardiovascular events in CAD-FluVac subjects.

Outcome (*n*)	CAD-FluVac (*n* = 137)		*p*-Value
	CV Event * (*n* = 34)	No CV Event (*n* = 103)	
Antibody A (Solomon Islands/3/2006 (H1N1))			
Protective (≥1:40) pre-vaccination, *n* (%)	25 (75.76%)	63 (66.32%)	0.39
Protective (≥1:40) post-vaccination, *n* (%)	33 (97.06%)	90 (94.74%)	0.33
Magnitude of change, ×fold, mean (SD)	97.61 (213.17)	44.14 (135.25)	0.65
Serologic response (≥4-fold rise), *n* (%)	24 (70.59%)	67 (65.05%)	1
Antibody B (Wisconsin/67/2005 (H3N2))			
Protective (≥1:40) pre-vaccination, *n* (%)	29 (87.88%)	87 (91.58%)	0.50
Protective (≥1:40) post-vaccination, *n* (%)	32 (96.97%)	93 (97.89%)	1
Magnitude of change, ×fold, mean (SD)	33.09 (122.06)	26.87 (112.74)	0.90
Serologic response (≥4-fold rise), *n* (%)	22 (66.67%)	61 (64.21%)	0.84
Antibody C (Malaysia/2506/2004)			
Protective (≥1:40) pre-vaccination, *n* (%)	29 (87.88%)	88 (92.63%)	0.47
Protective (≥1:40) post-vaccination, *n* (%)	33 (100.00%)	93 (97.89%)	1
Magnitude of change, ×fold, mean (SD)	9.27 (14.79)	13.55 (65.18)	0.56
Serologic response (≥4-fold rise), *n* (%)	26 (76.47%)	74 (71.84%)	1

* CV events: cardiovascular death, acute coronary syndrome, myocardial infarction, coronary artery bypass graft, or percutaneous intervention. CAD: coronary artery disease, IQR: Inter Quartile Range. Analysis: Fisher’s exact test used for protective level of antibody and serologic response and Kruskal–Wallis test used for magnitude of change.

## Data Availability

The data for this study are not publicly available, but they can be requested from the corresponding author.

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
