# Peer review of "Influenza Vaccination and Cardiovascular Outcomes in Patients with Coronary Artery Diseases: A Placebo-Controlled Randomized Study, IVCAD"

_vaccines, 2025, doi:10.3390/vaccines13050472_

Round 1

Reviewer 1 Report

Comments and Suggestions for Authors

Influenza Vaccination and Cardiovascular Outcomes in Patients with Coronary Artery Diseases, A Placebo-Controlled Randomized Study (IVCAD)

This paper addresses the potential protective effect of influenza vaccination in patients with coronary artery disease (CAD), which is important for the prevention of cardiovascular events. A randomized, placebo-controlled design increases the validity of the findings. Using logistic and Poisson regression provides a more precise estimate of the effects of vaccination on cardiovascular outcomes.

Suggestions for improvement:

  • The study included only 278 patients, which may be insufficient to detect statistically significant differences in mortality and cardiovascular events - a larger number of patients would ensure sufficient statistical power.
  • Lack of significant difference in mortality - The study showed no significant difference in overall mortality between the vaccinated and placebo groups (3.05% vs. 2.96%) - Longer-term studies (>24 months of follow-up) are needed to investigate the late effects of vaccination.
  • Unclear mechanism of protection - Although a slight trend toward a reduction in cardiovascular events was observed in vaccinated patients, it is not clear whether this effect is antibody-mediated, as serologic response did not correlate with outcomes. Non-antibody immune mechanisms, such as cytokine regulation, inflammatory markers (CRP, IL-6) and immunomodulatory effects of vaccination, need to be investigated.
  • Lack of data on seasonal effects - The study does not analyze how the effects of vaccination differ during the flu season vs. out of season. Patients should be followed during and outside the flu season, to clearly quantify the seasonal protective effect of vaccination.
  • Single vaccination (lack of data on revaccination) - Patients received only one dose of vaccine, and the effect of regular annual vaccination was not analyzed. Analysis of the effect of multiple vaccinations can provide better data on long-term protection.
  • Single measurement period for antibodies - Antibodies were measured only 1 month after vaccination, but it is not known how long the immune response lasts. Antibody measurements at 6, 12, and 24 months are required to determine long-term immune protection.
  • The study is single-blind, which increases the risk of researcher bias - Ideally, it would be double-blind to eliminate possible bias in outcome assessment.
Comments on the Quality of English Language

The English could be improved to more clearly express the research

Author Response

Response to reviewer 1 comments:

·       Comments 1:  The study included only 278 patients, which may be insufficient to detect statistically significant differences in mortality and cardiovascular events - a larger number of patients would ensure sufficient statistical power.

Response 1: We agree with the comment and appreciate your thoughtful insight. The study population is limited to 278 subjects, and this is a limitation that we have discussed as a shortcoming. This could have contributed to insignificant correlation despite showing a decrease in cardiovascular events. Unfortunately, we can’t change study population at this point. This has been proposed as a plan for future research in the conclusion section on page 10 as below:

“Further research, especially large double-blinded controlled clinical trials with longer follow-up period focused on non-antibody mediated correlates, is needed to elucidate the precise effect of vaccination on cardiovascular events and eliminate confounding effects.”

Comments 2: Lack of significant difference in mortality - The study showed no significant difference in overall mortality between the vaccinated and placebo groups (3.05% vs. 2.96%) - Longer-term studies (>24 months of follow-up) are needed to investigate the late effects of vaccination.

Response 2: We agree but this study could not be extended beyond one year due to lack of funding. We have listed this as a limitation of the study on page 10 and have proposed a longer study duration for future work.

·       Comments 3: Unclear mechanism of protection - Although a slight trend toward a reduction in cardiovascular events was observed in vaccinated patients, it is not clear whether this effect is antibody-mediated, as serologic response did not correlate with outcomes. Non-antibody immune mechanisms, such as cytokine regulation, inflammatory markers (CRP, IL-6) and immunomodulatory effects of vaccination, need to be investigated.

Response 3: This is absolutely needed, and we do agree. Additional discussion was added to the page 9, as below:

“IVCAD study also did not demonstrate a significant difference in antibody response to influenza vaccine in CAD patients in subpopulations with and without cardiovascular events. In other words, antibody response does not seem to be the only mechanism involved in influenza vaccine effect on CV outcome. To the best of our knowledge there is no other existing parallel data investigating CV outcomes based on antibody response to influenza vaccine. In a prior publication in IVCAD study population, antibody response in CAD patients was comparable to healthy controls [17]. A recent similar clinical trial investigated antibody response to high-dose trivalent compared to standard quadrivalent influenza vaccine in subjects with heart failure or myocardial infarction. A more robust humoral response was observed to high-dose vaccine, but antibody response had no association with mortality or cardiopulmonary hospitalization [42].

Inflammation is well established to be one of the main baseline conditions resulting in CV events and multiple studies showed protective correlation of influenza vaccine by preventing influenza infection and reducing systemic inflammation [11, 39, 40]. So, vaccine has a direct protective effect by preventing influenza infection primarily via antibody response [41]. On the other hand, influenza vaccine has been shown to have non-influenza season reduction in cardiovascular mortality [43,44] which could not be well investigated in this trial due to small sample size. These findings suggest non-antibody related or nonspecific effect of influenza vaccine on reducing cardiovascular events. Immune system modulation is a possible explanation. Since a regulated immune system is associated with cardiovascular health, influenza vaccine plays the role with modification of plaque instability and pre/post infarction inflammation [45-48]. Moreover, cellular immune response following influenza vaccine might have a more important role compared to humoral immunity [49,50]. Although exact mechanisms and pathways require further investigations, preliminary data supports the idea that future research should focus on parallel pathways including cellular immunity and cytokines regulations in response to influenza vaccination to identify this correlation.”

·       Comments 4: Lack of data on seasonal effects - The study does not analyze how the effects of vaccination differ during the flu season vs. out of season. Patients should be followed during and outside the flu season, to clearly quantify the seasonal protective effect of vaccination.

Response 4: Unfortunately, number of outcome events were so low that we could not implement seasonal effect. This requires a larger study population with longer duration of follow up which was not possible within current study due to lack of funding. We mentioned that in limitation part in page 10. 

·       Comment 5: Single vaccination (lack of data on revaccination) - Patients received only one dose of vaccine, and the effect of regular annual vaccination was not analyzed. Analysis of the effect of multiple vaccinations can provide better data on long-term protection.

Response 5: We are totally in agreement. The study was covered under a grant for only one-year, and this is obviously a limitation. We mentioned that in limitation part in page 10.

·       Comments 6. Single measurement period for antibodies - Antibodies were measured only 1 month after vaccination, but it is not known how long the immune response lasts. Antibody measurements at 6, 12, and 24 months are required to determine long-term immune protection.

Response 6: We appreciate this comment, though we don’t believe the cardiovascular protection is antibody related. This was not even significant for one-month antibody level.

Moreover, most subjects were vaccinated in fall and winter, so antibody titers beyond 3-6 months were out of influenza peak season and even having protective antibody titer outside that window would not be valuable.

As a result, we believe that measurement of antibody beyond 1 months would have not added any additional information.

·       Comments 7. The study is single-blind, which increases the risk of researcher bias. Ideally, it would be double-blind to eliminate possible bias in outcome assessment.

Response 7: We totally agree but double blind was technically difficult, and we could not implement this due to difficulty in obtaining matched placebo from the manufacturing company. We have mentioned that in limitation section on page 10.

Comments 8 on the Quality of English Language: The English could be improved to more clearly express the research

Response 8: We appreciate your comment. To refine English, we requested the language edition form MDPI.

Reviewer 2 Report

Comments and Suggestions for Authors

This single-blind, randomized, placebo-controlled trial examines the effect of influenza vaccination on cardiovascular outcomes in patients with coronary artery disease (CAD). The study reports a non-significant reduction in mortality but suggests a trend toward fewer non-fatal cardiovascular events (e.g., acute coronary syndrome [ACS], myocardial infarction [MI]) in the vaccinated group. While the study offers valuable insights into the potential cardioprotective role of influenza vaccination and is both interesting and meaningful, and I have only a few comments before the paper can be accepted.

1. Line 40: The authors indicate that influenza vaccination may improve cardiovascular outcomes. However, primary (mortality) and secondary outcomes (non-fatal events) were non-significant.

2. Line 75: The study enrolled 278 patients (target: 300), and non-significant trends in secondary outcomes (e.g., ACS, MI) may reflect underpowered analyses.

3. Line 76: Explore effect heterogeneity by age (≥65 years), diabetes status, or baseline Gensini score.

4. Line 201: Despite robust antibody responses (Table 3), no correlation with cardiovascular outcomes was observed (Table 4). This paradox is insufficiently explored. It is better to expand the discussion to include non-antibody mechanisms (e.g., trained immunity, anti-inflammatory effects) and cite relevant literature.

5. Line 217: Discuss cost-effectiveness or real-world implementation challenges for CAD populations.

6. Line 259: The discussion lacks integration with recent evidence on vaccine-induced immunomodulation. Recommende strengthen mechanistic hypotheses by linking findings to broader immunological concepts.

7. Line 265: The discussion lacks integration with recent evidence on vaccine-induced immunomodulation. Recommende strengthen mechanistic hypotheses by linking findings to broader immunological concepts.

8. Line 307: Duplicate entries (Refs 13–14) and missing DOIs/PMIDs require correction.

9. Line 340 and Line342: Duplicate entries.

Author Response

Response to reviewer 2 comments:

Comments 1:  Line 40: The authors indicate that influenza vaccination may improve cardiovascular outcomes. However, primary (mortality) and secondary outcomes (non-fatal events) were non-significant.

Response 1: Influenza vaccination was associated with a 27% reduction in the odds of cardiovascular adverse events (ACS, MI, cardiovascular interventions), although this did not reach statistical significance. We understand that this might cause confusion. We believe this could be related to limited sample size and short duration of follow up which are limitations of our study. This has been discussed in section 3.2. 12-months outcome.

Comments 2: Line 75: The study enrolled 278 patients (target: 300), and non-significant trends in secondary outcomes (e.g., ACS, MI) may reflect underpowered analyses.

Response 2: We agree with the reviewer that our analyses may have been underpowered to detect differences in our secondary outcomes. We have added the following statement to discussion section on page 10 as below:

“Our sample size was calculated to have 70% power to detect a 10% difference in protection rate between treatment arms at α=0.05, and may have been underpowered to detect differences in secondary outcomes”

Comments 3: Line 76: Explore effect heterogeneity by age (≥65 years), diabetes status, or baseline Gensini score.

Response 3: We appreciate the reviewer’s recommendation to explore effect heterogeneity by age, diabetes status or baseline Gensini score. We also agree that such exploration of heterogeneity would be of interest to the clinical and research community. However, given our sample size and lack of a significant main effect between treatment arms we do not think exploration of treatment effect heterogeneity is justifiable. Further to this point, our primary outcome of cardiovascular mortality occurred in 6 patients. Thus, although our sample size was 278, our effective sample size for assessment of our primary outcome was 6 events. Stratifying these 6 events would not provide sufficient data and power to explore heterogeneity in our study.

Comments 4: Line 201: Despite robust antibody responses (Table 3), no correlation with cardiovascular outcomes was observed (Table 4). This paradox is insufficiently explored. It is better to expand the discussion to include non-antibody mechanisms (e.g., trained immunity, anti-inflammatory effects) and cite relevant literature.

Response 4: This is absolutely needed, and we do agree. Additional discussion was added on page 9 as below:

“IVCAD study also did not demonstrate a significant difference in antibody response to influenza vaccine in CAD patients in subpopulations with and without cardiovascular events. In other words, antibody response does not seem to be the only mechanism involved in influenza vaccine effect on CV outcome. To the best of our knowledge there is no other existing parallel data investigating CV outcomes based on antibody response to influenza vaccine. In a prior publication in IVCAD study population, antibody response in CAD patients was comparable to healthy controls [17]. A recent similar clinical trial investigated antibody response to high-dose trivalent compared to standard quadrivalent influenza vaccine in subjects with heart failure or myocardial infarction. A more robust humoral response was observed to high-dose vaccine, but antibody response had no association with mortality or cardiopulmonary hospitalization [42].

Inflammation is well established to be one of the main baseline conditions resulting in CV events and multiple studies showed protective correlation of influenza vaccine by preventing influenza infection and reducing systemic inflammation [11, 39, 40]. So, vaccine has a direct protective effect by preventing influenza infection primarily via antibody response [41]. On the other hand, influenza vaccine has been shown to have non-influenza season reduction in cardiovascular mortality [43,44] which could not be well investigated in this trial due to small sample size. These findings suggest non-antibody related or nonspecific effect of influenza vaccine on reducing cardiovascular events. Immune system modulation is a possible explanation. Since a regulated immune system is associated with cardiovascular health, influenza vaccine plays the role with modification of plaque instability and pre/post infarction inflammation [45-48]. Moreover, cellular immune response following influenza vaccine might have a more important role compared to humoral immunity [49,50]. Although exact mechanisms and pathways require further investigations, preliminary data supports the idea that future research should focus on parallel pathways including cellular immunity and cytokines regulations in response to influenza vaccination to identify this correlation.”

Comment 5: Line 217: Discuss cost-effectiveness or real-world implementation challenges for CAD populations

Response 5: We have addressed this in discussion and added related references in page 9 as below:

“Influenza is a serious illness which can impact high-risk population causing high mortality and morbidity with subsequent significant impact on healthcare resources. This is more prominent in elderly and individuals with chronic diseases. Influenza infection has higher impact on CVD patients and can lead to MI, HF leading to hospitalizations and readmissions due to subsequent morbidities.”

Comments 6. Line 259: The discussion lacks integration with recent evidence on vaccine-induced immunomodulation. Recommend strengthen mechanistic hypotheses by linking findings to broader immunological concepts.

Response 6: We have addressed this in discussion and additional discussion added to the page 9 (see response to comment 4).

Comments 7. Line 265: The discussion lacks integration with recent evidence on vaccine-induced immunomodulation. Recommend strengthen mechanistic hypotheses by linking findings to broader immunological concepts.

Response 7: We have addressed this in discussion and additional discussion was added to the page 9 (see response to comment 4).

Comments 8. Line 307: Duplicate entries (Refs 13–14) and missing DOIs/PMIDs require correction.

Response 8: Duplicated reference removed.

Comments 9. Line 340 and Line342: Duplicate entries.

Response 9: Duplicated reference removed.

Reviewer 3 Report

Comments and Suggestions for Authors

Review comments on vaccines-3523271

Journal: Vaccines

Manuscript ID: vaccines-3523271

Title: Influenza Vaccination and Cardiovascular Outcomes in Patients with Coronary Artery Diseases, A placebo-controlled randomized study; IVCAD

Authors: Mohammadmoein Dehesh, Sharareh Gholamin, Seyed-Mostafa Razavi, Ali Skandari, Hossein Vakili, Mohammad Rahnavardi Azari, Yunzhi Wang, Ethan Gough, Maryam Keshtkar-Jahromi *

Influenza Virus Vaccines

Special Issue: The Recent Development of Influenza Vaccine: 2nd Edition

Major comments:

  1. This is the phase 2 report on the single blinded clinical trial for evaluating the correlation between influenza vaccine (FluVac) and cardiovascular outcomes in Coronary Artery Disease (CAD) patients, although actual clinical trial was conducted 2008~2009 in Tehran, Iran. Authors published their phase 1 report previously in 2009 (cited as ref. 17). In the present report, authors evaluated cardiovascular outcomes 12 months after the vaccination in vaccine recipients in comparison with placebo controls.
  2. In the present report, authors showed that influenza vaccination was associated with lower cardiovascular events in CAD population. However, antibody titer was not correlated with cardiovascular outcomes. Therefore, authors explained these observations as due to non-antibody mediated molecular and cellular immune regulatory mechanisms to be operative.
  3. These observations are very interesting in considering the controversial issues between influenza vaccination and cardiovascular outocomes. Accordingly, this manuscript may be potentially acceptable for publication in “Vaccines” special issue. However, before the final decision to be made, following points should be clarified.
  4. In the introduction section, authors described that “In the current trial, we evaluated correlation between 2007-2008 influenza vaccine (FluVac) and cardiovascular outcomes in CAD patients. In phase 1 (previously reported), we evaluated antibody (Ab) response to influenza vaccine in CAD patients which was comparable to healthy controls [17]. In phase 2 (current report), we evaluated cardiovascular outcomes 12 months after vaccination in vaccine recipients in comparison with placebo controls. The delay in reporting the result was nothing other than our time limitations. (page 2, lines 58-64)”. Their previous report (phase 1 of their clinical trial) corresponds to ref. 17 (published in 2010), in which the first author “Mohammadmoein Dehesh” of current study was not included as coauthors. On the other hand, the roles of this first author in the current study are indicated as “validation” and “writing”. Accordingly, I am not sure about the validity of the first author. Corresponding author should make some statements for clarification.
  5. In Table 1(page 5), features and background history of participants are tabulated. However, for the column of “CAD-FluVac (n=137)”, the values for “Hypertension n (%)” are indicated as 115 (83.9), which are very different from the corresponding values for “Hypertension n (%)” in Table 1 of ref.17. For other features, such as age, male/female, diabetes mellitus, hyperlipidemia, identical values are found, respectively. Why?
  6. In Table 2 (page 5); what is the definition of “n (%)”? Usually, that should be “number and percentage”. However, percentages do not in agreement with the total number indicated at the top line. Based on my calculation, top line should be as follows. All (n=180), CAD-FluVac (n=88), CAD-Placebo (n=92).

However, in the lower parts of the same Table 2, n (%) are consistent with the numbers indicated in the top line; i.e., All (n=278), CAD-FluVac (n=137), CAD-Placebo (n=141).

  1. Similarly, in Tables 3 and 4 (page 7); The values “n (%)” for respective line are inconsistent with the total values indicated at the top line in each Table.

Minor comments:

(Page 1, line 33) “subejcts” should be “subjects”.

(Page 7, Table 4) A line under “Antibody A (Solomon Islands/3/2006(H1N1)”should be deleted.

(Page 7, Table 3) Top line in the table, “CAD-Placebo (n=137)” should be “CAD-Placebo (n=141)”. Similarly, in the same line, “CAD-FluVac (n=141)” should be “CAD-FluVac (n=137)”.

Author Response

Response to reviewer 3 comments:

Comments 1: Their previous report (phase 1 of their clinical trial) corresponds to ref. 17 (published in 2010), in which the first author “Mohammadmoein Dehesh” of current study was not included as coauthors. On the other hand, the roles of this first author in the current study are indicated as “validation” and “writing”. Accordingly, I am not sure about the validity of the first author. Corresponding author should make some statements for clarification.

Response 1: Thank you for pointing this out. We appreciate your careful review. For further clarification, reference 17 is another study with a similar population (Vaccinated CAD subjects compared to healthy adults). Aims of these two studies as well as subject populations are different. Although most of team members are involved with both studies, 3 authors including Dr Dehesh contributed only to the current study. Dr Dehesh contributed to defining objectives, data collection, data validation, statistical analysis, interpreting results, writing the paper, journal submission and all communications within the study team. We believe that he well deserves to be the first author based on his significant contribution to the work. 

Comments 2: In Table 1(page 5), features and background history of participants are tabulated. However, for the column of “CAD-FluVac (n=137)”, the values for “Hypertension n (%)” are indicated as 115 (83.9), which are very different from the corresponding values for “Hypertension n (%)” in Table 1 of ref.17. For other features, such as age, male/female, diabetes mellitus, hyperlipidemia, identical values are found, respectively. why?

Response 2: We agree that Hypertension values should be the same as these are the same CAD populations with the same number of subjects. We have double checked our data and the numbers listed in this paper are correct. We believe for some reason the numbers in prior publication were listed in error and was not picked up in error.

Comments 3: In Table 2 (page 5); what is the definition of “n (%)”? Usually, that should be “number and percentage”. However, percentages do not in agreement with the total number indicated at the top line. Based on my calculation, top line should be as follows. All (n=180), CAD-FluVac (n=88), CAD-Placebo (n=92). However, in the lower parts of the same Table 2, n (%) are consistent with the numbers indicated in the top line; i.e., All (n=278), CAD-FluVac (n=137), CAD-Placebo (n=141). Similarly, in Tables 3 and 4 (page 7); The values “n (%)” for respective line are inconsistent with the total values indicated at the top line in each Table

Response 3: Agree. We have accordingly changed the number in the table2. The reason for this discrepancy was that not all the patients have echocardiogram but all of them had the questionnaire filled out. We added reference numbers for each line in the table 2.

Minor comments:

(Page 1, line 33) “subejcts” should be “subjects”.

Response: We have corrected this.

(Page 7, Table 4) A line under “Antibody A (Solomon Islands/3/2006(H1N1)” should be deleted.

Response: It seems there is not a line in the word file, could be due to some changes in the version provided to you. We will make sure it is corrected in the final draft.

(Page 7, Table 3) Top line in the table, “CAD-Placebo (n=137)” should be “CAD-Placebo (n=141)”. Similarly, in the same line, “CAD-FluVac (n=141)” should be “CAD-FluVac (n=137)”

Response: We have corrected this error.

Round 2

Reviewer 1 Report

Comments and Suggestions for Authors

Overall, the authors adequately addressed the comments and acknowledged the study's limitations. However, some aspects could have been better addressed:

The authors acknowledged that the sample size and statistical power were limited but did not discuss the statistical power of the analysis (e.g., whether it was sufficient to detect clinically relevant differences). It might have been useful to include a brief discussion of this.

When it comes to seasonal effects, the response is valid, but you could have added a reference to support the claim that the numbers are too small for this kind of analysis.

The authors also acknowledge that they could not have extended the study but could have added literature context (e.g., existing studies that were longer in duration) to further support their limitations.

Author Response

Response to reviewer comments round 2:

Comments 1:  The authors acknowledged that the sample size and statistical power were limited but did not discuss the statistical power of the analysis (e.g., whether it was sufficient to detect clinically relevant differences). It might have been useful to include a brief discussion of this.

Response 1: We agree with the comment and appreciate your thoughtful insight. We addressed that in the limitation part with more clear explanation about study sample size and event rate.

The revised text in page 10 as below:

“Our sample size was calculated to have 70% power to detect a 10% difference in protection rate between treatment arms at α=0.05 [7] and may have been underpowered to detect differences in outcomes due the low outcome event rates in the placebo arm in our study sample (<10% for most outcomes). Given the low event rates we observed, absolute risk differences between treatment arms were generally <=2%, and were thus below our target effect size. Larger trials are needed to determine more precise estimate of effect for this vaccine.”

Comments 2: When it comes to seasonal effects, the response is valid, but you could have added a reference to support the claim that the numbers are too small for this kind of analysis.

Response 2: We agree, and we added an explanation to page 10 that due to low outcome rate subgroup analysis was not feasible.

“In contrast to some more recently designed trials, we did not record the seasonal outcomes, which might have provided more detailed and related cardiovascular outcomes during peak virus circulation. Also, due to low number of outcome events, a reliable subgroup analysis to explore seasonal variations was not feasible [59].”

Comments 3: The authors also acknowledge that they could not have extended the study but could have added literature context (e.g., existing studies that were longer in duration) to further support their limitations.

Response 3: This was needed, and we do agree. Additional discussion was added to the page 10, as below:

“The follow-up period for the current study was 12 months, which is similar to most of the other studies, but a longer follow-up period with serial annual vaccinations would have been preferred. Moreover, the outcomes were cumulative and could have happened any time after vaccination until 12 months postvaccination. A few clinical trials, such as the study by Loeb in 2022, which followed subjects for 2.3 years with annual vaccination [15], and the study by Modin in 2018, with a follow-up period of 3.7 years and variable vaccine coverage [32], reported similar results demonstrating the cardiovascular protective effects of the influenza vaccine.”